# CONCUR: A FRAMEWORK FOR CONTINUAL CONSTRAINED AND UNCONSTRAINED ROUTING

**Peter Baile Chen**[1]    **Weiyue Li**[2]    **Dan Roth**[3]
**Michael Cafarella**[1]    **Samuel Madden**[1]    **Jacob Andreas**[1]
[1]MIT    [2]Harvard University    [3]Oracle AI and the University of Pennsylvania
Correspondence: `peterbc@mit.edu`

## ABSTRACT

AI tasks differ in complexity and are best addressed with different computation strategies (e.g., combinations of models and decoding methods). Hence, an effective routing system that maps tasks to the appropriate strategies is crucial. Most prior methods build the routing framework by training a *single* model across *all* strategies, which demands full retraining whenever new strategies appear and leads to high overhead. Attempts at such continual routing, however, often face difficulties with generalization. Prior models also typically use a *single* input representation, limiting their ability to capture the full complexity of the routing problem and leading to sub-optimal routing decisions. To address these gaps, we propose CONCUR, a **con**tinual routing framework that supports both **c**onstrained and **u**nconstrained **r**outing (i.e., routing with or without a budget). Our *modular* design trains a separate predictor model for each strategy, enabling seamless incorporation of new strategies with low additional training cost. Our predictors also leverage *multiple* representations of both tasks and computation strategies to better capture overall problem complexity. Experiments on both in-distribution and out-of-distribution, knowledge- and reasoning-intensive tasks show that our method outperforms the best single strategy and strong existing routing techniques with higher end-to-end accuracy and lower inference cost in both continual and non-continual settings, while also reducing training cost in the continual setting.[1]

## 1 INTRODUCTION

AI tasks vary in difficulty, and thus are optimally served by different computation strategies, such as selecting appropriate models (small or large language models) and decoding methods (with or without chain-of-thought reasoning (Wei et al., 2022)). Effective routing ensures tasks are paired with the most suitable strategies to help improve overall accuracy and reduce runtime and costs.

Prior routing work (Zhu et al., 2025; Ong et al., 2024; Ding et al., 2024; Lu et al., 2024; Hari & Thomson, 2023; Chen et al., 2024; Feng et al., 2025; Pan et al., 2025; Zhuang et al., 2024; Liu et al., 2024; Sakota et al., 2024; Mohammadshahi et al., 2024; Nguyen et al., 2024; Damani et al., 2024) typically employs a fixed set of computation strategies, relying on a *single* model trained jointly on data from *all* strategies. However, this monolithic design limits generalization to *continual* settings where routers need to quickly adapt to previously unseen strategies. In practice, continual routing is crucial, as better and increasingly efficient models and decoding methods are constantly emerging, and not incorporating them promptly risks missing potential gains in accuracy and reductions in computational cost. However, whenever a novel strategy appears, existing approaches require retraining the model from scratch using data that covers both previous and new strategies.

Although some recent efforts attempt to move toward a continual setting, they raise significant concerns about generalizability. For example, Wang et al. (2025) adopts a modular design by training separate router models for different computation strategies. However, since these router architectures differ and are specifically tailored to individual strategies, extending them to unseen strategies remains non-trivial. Jitkrittum et al. (2025) introduces a zero-shot router based on model feature vec-

---

[1]Data and code are available at `https://peterbaile.github.io/concur/`.

Figure 1: CONCUR learns one predictor per computation strategy that uses multiple input representations to support continual routing and better routing decisions under both continual and non-continual settings.

tors, enabling generalization to unseen models without retraining. However, their method depends on *predefined* prompts, which limits its adaptability to varied prompts and tasks.

Besides efficiency concerns in the continual setting, prior work also raises concerns about end-to-end performance (accuracy and inference cost) in both continual and non-continual settings. In principle, models should adopt flexible parameterizations that allow them to combine both general-purpose and task- or strategy-specific signals, thereby capturing richer information about the routing problem and enabling high-quality routing decisions. However, prior work typically relies on highly restricted parameterizations; for example, Ong et al. (2024) and Zhu et al. (2025) parameterize only task-level representations, while Zhuang et al. (2024) and Pan et al. (2025) restrict themselves to a *single* parameterization of computation strategies and input tasks, respectively. Such limited designs may reduce expressivity and constrain the quality of routing decisions.

Motivated by these issues, we propose a generalizable routing framework applicable to both continual and non-continual, as well as constrained and unconstrained, settings, as illustrated in Figure 1. Our framework adopts a *modular* design, where *separate* predictor models are trained for each computation strategy, using *both* general-purpose and task-specific representations of input tasks and computation strategies to estimate accuracy and efficiency. These estimates are then used to formulate constrained and unconstrained routing as optimization problems, which can subsequently be solved to determine the optimal routing decisions.

In contrast to prior approaches that rely on a single model trained on data from all strategies, our modular predictor design makes continual routing far more practical. New strategies can be incorporated simply by training an additional predictor, without retraining existing ones, thus avoiding costly overhead. Moreover, unlike prior continual routing efforts that lack generalizability, either by tailoring router architectures to specific strategies or by relying on fixed prompts, our predictors share the same model architectures, and our method imposes no restrictions on prompts or task diversity.

In addition, rather than restricting to a single representation as in prior work, our architecture incorporates multiple representations of both input tasks and computation strategies to capture richer information about the routing problem, enabling more accurate routing decisions and improved end-to-end performance in both continual and non-continual settings.

In summary, we present CONCUR, a framework for **con**tinual **c**onstrained and **u**nconstrained **r**outing. CONCUR trains modular predictors for accuracy and efficiency that draw on both general-purpose and task-specific representations of input tasks and computation strategies, and integrates these estimates with specific routing algorithms to address constrained and unconstrained routing. We evaluated CONCUR on diverse benchmarks (including multi-hop QA, general reasoning multiple-choice tasks, and math problems) across both in- and out-of-distribution settings. Results show that CONCUR consistently outperforms the best single strategy baseline and existing routing methods, achieving higher end-to-end accuracy and lower inference cost in both continual and non-continual settings, as well as improved training efficiency in the continual setting.

## 2 METHODOLOGY

As outlined in Section 1, our goal is to build a routing framework that supports continual settings and improves end-to-end performance in both continual and non-continual settings. The core ideas be-

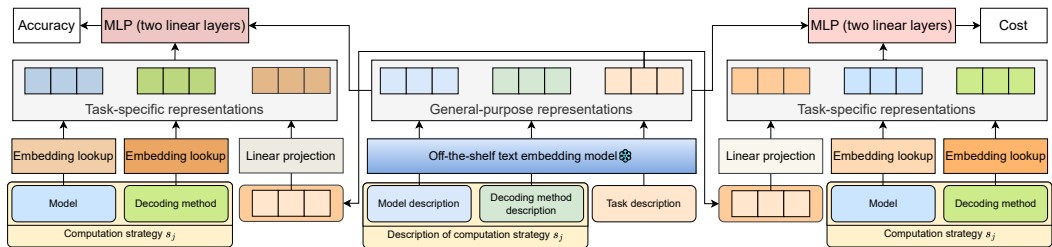

Figure 2: Overall predictor architecture of CONCUR. For each computation strategy $s_j$, we train two predictor models: one estimates the accuracy of applying $s_j$ to the input task, and the other estimates its cost, using both general-purpose and task-specific representations.

hind our routing models are: (1) we adopt a modular design, training a separate model for each strategy so that extending to new strategies only requires training additional predictors without touching existing models; and (2) we use multiple input representations to better capture the complexity of the routing problem, rather than relying on a single representation.

Concretely, we formulate both constrained and unconstrained routing as optimization problems over accuracy and efficiency. Therefore, predictors are trained to estimate these metrics, which are subsequently used to solve the optimization problems. Building on this design, Section 2.1 describes how we train modular predictors to model task difficulty using multiple input representations, and Section 2.2 explains how these predictions drive routing decisions.

## 2.1 PREDICTORS

We outline the training of predictors designed to estimate the performance of applying a computation strategy to a given user task. This involves characterizing input tasks and strategies, detailing the predictor model architectures, and explaining the training procedure and prediction process.

**Characterization.** We define a task $t_i$ as comprising both the question and any related context. While prior work (Ong et al., 2024; Zhuang et al., 2024; Pan et al., 2025) typically considers only the question, a task may also include supporting documents, such as in the open-domain QA setting, that provide useful signals for estimating task difficulty. We define a computation strategy $s_j$ as a (model, decoding method) pair $(m_j, d_j)$. In our implementation, a model denotes the underlying language model (e.g., Qwen2.5-7B-Instruct), while a decoding method refers to the decoding algorithm (e.g., chain-of-thought). However, developers can broaden the definition of computation strategies to incorporate additional parameters. Given a set of supported language models $M$ and decoding methods $D$, the complete set of computation strategies $S$ is the Cartesian product $S = M \times D$.

The performance of applying strategy $s_j$ to task $t_i$ includes both the accuracy $a_{ij}$ and the inference cost $c_{ij}$, which we use as a proxy for efficiency. In our setting, we measure the cost using FLOPs. FLOPs are preferred over token counts because model sizes vary, and the computational cost of generating a single token differs across models. Using FLOPs allows for a more standardized comparison of efficiency across heterogeneous models. However, depending on the deployment environment (e.g., whether models are quantized), FLOPs can be replaced by other numerical efficiency metrics, including throughput, latency, or energy consumption.

**Architecture.** To achieve high-quality predictions, we train two independently parameterized predictors: one for estimating accuracy and the other for estimating cost. Each predictor incorporates both general-purpose and task-specific representations of the input task $t_i$ and strategy $s_j = (m_j, d_j)$, enabling them to capture both general and task-strategy specific characteristics. For every $s_j$, two such predictors are trained, resulting in a *modular* design where predictors for different strategies can be trained independently. This allows new strategies to be supported by training only the corresponding predictors, leaving existing ones untouched and incurring minimal overhead. The overall architecture is shown in Figure 2.

*1. General-purpose representation.* We generate a general-purpose representation by passing the textual description of the task and strategy (textual descriptions of the strategies are provided in

Appendix A) through an off-the-shelf text embedding model $R$.

$$\mathbf{g}_i^t = R(t_i), \; \mathbf{g}_j^m = R(m_j), \; \mathbf{g}_j^d = R(d_j)$$

Concatenating the three parts gives the general representation $\mathbf{g}_{ij} = [\mathbf{g}_i^t; \mathbf{g}_j^m; \mathbf{g}_j^d] \in \mathbb{R}^{3k}$ where $k$ is the dimension of the encoded representation.

*2. Task-specific representation.* Task-specific representations are derived using learnable projections and embeddings. The task-specific representation of the input task is derived by linearly projecting its previously defined general-purpose representation. Task-specific representations for the model and decoding method are derived from learned embeddings that map model and decoding method IDs into trainable dense vectors optimized alongside the rest of the model.

$$\mathbf{t}_i^a = W_t^a R(t_i), \; \mathbf{m}_j^a = E_M^a[m_j], \; \mathbf{d}_j^a = E_D^a[d_j]; \; \mathbf{t}_i^c = W_t^c R(t_i), \; \mathbf{m}_j^c = E_M^c[m_j], \; \mathbf{d}_j^c = E_D^c[d_j]$$

where $X^a$ and $X^c$ represent $X$ in the accuracy and cost predictor models, respectively. The metric (i.e., accuracy or cost)-specific linear projection is denoted by $W_t \in \mathbb{R}^{k \times k}$, while $E_M$ and $E_D$ are metric-specific embedding lookup tables for models and decoding methods, respectively.

Concatenating the three parts gives the task-specific representations $\mathbf{s}_{ij}^a = [\mathbf{t}_i^a; \mathbf{m}_j^a; \mathbf{d}_j^a]$ and $\mathbf{s}_{ij}^c = [\mathbf{t}_i^c; \mathbf{m}_j^c; \mathbf{d}_j^c] \in \mathbb{R}^{3k}$.

*3. MLP.* Finally, we concatenate the general-purpose and task-specific representations and feed them through two linear layers to produce the accuracy and cost predictions, $\hat{a}_{ij}$ and $\hat{c}_{ij}$.

$$\hat{a}_{ij} = f^a([\mathbf{g}_{ij}; \mathbf{s}_{ij}^a]); \; \hat{c}_{ij} = f^c([\mathbf{g}_{ij}; \mathbf{s}_{ij}^c])$$

where $f^a$ is a binary classifier and $f^c$ is a regressor for predicting the accuracy and cost, respectively.

We note that once trained, the representations of a strategy $s_j$ remain fixed and thus contribute a constant term to the predictions. Nonetheless, as we will show in Section 3, including these strategy representations improves performance over strong routing baselines.

**Training.** Using the training tasks and their target answers, we apply each strategy $s_j$ to each task $t_i$ to obtain the ground-truth labels $a_{ij}$ (by comparing the generated and the target answers) and $c_{ij}$. Due to the modular design with respect to each $s_j$, training a predictor for a given strategy *only* requires the training data associated with that strategy. The training procedure for the predictors is as follows. The accuracy predictor is a binary classifier trained with cross-entropy loss:

$$L_{acc} = -a_{ij} \log(\hat{a}_{ij}) - (1 - a_{ij}) \log(1 - \hat{a}_{ij})$$

where $a_{ij}$ is the ground-truth accuracy label and $\hat{a}_{ij}$ is its predicted value. The cost predictor is a regressor trained with mean squared error loss:

$$L_{cost} = (c_{ij} - \hat{c}_{ij})^2$$

where $c_{ij}$ is the ground-truth cost and $\hat{c}_{ij}$ is its predicted value.

**Inference.** For a new task $t_i$, we encode it with the same embedding model used during training and pass this representation, along with the representation of each strategy $s_j$, through the respective accuracy and cost predictors to obtain $\hat{a}_{ij}$ and $\hat{c}_{ij}$.

**Continual routing.** Continual routing refers to a setting where the routing framework is trained on an initial set of strategies $S$ and must later incorporate additional strategies $S'$ that become available at a later time. Our modular design assigns a separate predictor to each strategy $s_j$, so incorporating a new strategy $s_j'$ involves training only its predictors, leaving previously trained models unchanged, making extensions straightforward and efficient. Naturally, our framework also applies to the non-continual setting where the set of strategies remains fixed over time.

## 2.2 ROUTING

Using the predicted accuracy $a_{ij}$ and cost $c_{ij}$ of applying strategy $s_j$ to task $t_i$, we demonstrate how both constrained and unconstrained routing can be formulated as optimization problems and solved, leading to the final routing decisions.

**Unconstrained routing.** For a given task $t_i$, unconstrained routing involves choosing the computation strategy that achieves an optimal trade-off between accuracy and cost. This can be framed as the following bi-objective optimization problem, maximizing accuracy while simultaneously minimizing cost: $\max_j a_{ij}, \min_j c_{ij}$.

By introducing a weight $w$ to represent the trade-off between accuracy and cost, the bi-objective optimization problem can be reformulated as a single-objective problem that maximizes the weighted sum of these two objectives.

$$\max_j \sum_i (w \cdot a_{ij} + (1 - w) \cdot (-c_{ij})) = \sum_i \max_j (w \cdot a_{ij} + (1 - w) \cdot (-c_{ij})) \quad (1)$$

Then, $t_i$ is routed to the strategy $s_j^*$ that maximizes the weighted sum.

**Constrained routing.** For a task $t_i$ with an associated cost budget $B$, constrained routing seeks the computation strategy that delivers the highest possible accuracy without exceeding the budget. This can be expressed as the following optimization problem: $\max_{j, c_{ij} \leq B} a_{ij}$

However, optimizing each task individually may not yield the best overall result, as local optima do not always translate to global optimality. For a batch of $n$ tasks $t_1, t_2, ..., t_n$, the constrained optimization problem can be reformulated as

$$\max_{j, \sum_i c_{ij} \leq nB} \sum_i a_{ij} \quad (2)$$

We address this optimization problem by formulating a dynamic programming (DP) approach (see Appendix B for details), which has an overall complexity of $O(n \cdot nB \cdot |S|) = O(n^2 \cdot B|S|)$, where $B$ represents the budget per task and $|S|$ is the number of computation strategies. Since the budget can be kept relatively small through scaling, and the number of computation strategies (i.e., LLM-decoding pairs) used simultaneously is usually moderate, the DP algorithm can be solved efficiently for a reasonable number of tasks.

Although the above formulation is designed to maximize accuracy under a cost constraint, it can be straightforwardly adapted to minimize cost for a given accuracy requirement.

## 3 EXPERIMENTS

### 3.1 EXPERIMENTAL SETUP

**Datasets.** We select a diverse set of tasks covering different skills and output formats: factual multi-hop question answering with short text answers, general reasoning with multiple-choice answers, and mathematical problems requiring numerical answers. For each task category, we select two datasets: one in-distribution for both training and testing, and one out-of-distribution reserved exclusively for testing. A summary of these datasets and their statistics is provided in Table 1.

Table 1: Datasets and their sizes.

| | In-distribution | | Out-of-distribution | |
|---|---|---|---|---|
| | Dataset | Test size | Dataset | Test size |
| Multi-hop QA | 2WikiMultiHop (Ho et al., 2020) | 1000 | HotpotQA (Yang et al., 2018) | 500 |
| General reasoning | MMLU (Hendrycks et al., 2021) | 1000 | GPQA (Rein et al., 2024) | 448 |
| Math problems | GSM8k (Cobbe et al., 2021) | 1000 | SVAMP (Patel et al., 2021) | 500 |

**Computation strategies.** As described in Section 2.1, the set of computation strategies comprises combinations of models and decoding methods. For models, we used Qwen2.5-Instruct models (1.5B, 3B, and 7B) and Llama-3.x-Instruct models (3.2-3B and 3.1-8B). For decoding, we considered two common approaches: *vanilla*, where models directly generate the answer, and *chain-of-thought* (Wei et al., 2022), where models produce intermediate reasoning steps before the final answer. The descriptions and prompts for each strategy are presented in Appendix A and Appendix C, respectively. In total, this yields five LLMs and two decoding methods, for a total of ten strategies.

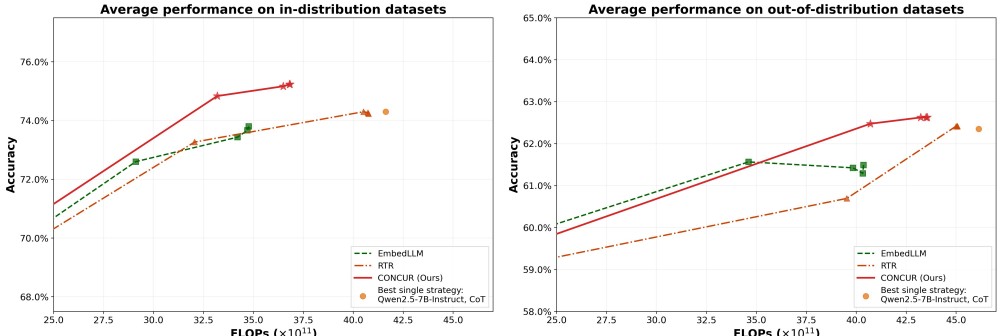

Figure 3: Pareto curves for unconstrained routing on both in- and out-of-distribution datasets across various values of $w$ defined in Section 2.2, illustrating the trade-off between accuracy and cost. Full diagrams are available in Appendix E.

**Baselines.** We compare our routing framework against the best single strategy without routing and three strong routing baselines. Implementation details are provided in Appendix D.

(1) Best single strategy: We use the same model-decoding pair that achieves the highest overall accuracy across all tasks. In this case, Qwen2.5-7B-Instruct with chain-of-thought (CoT) decoding is selected for its superior accuracy.

(2) RouteLLM (Ong et al., 2024): This approach uses a *single* classifier that relies on the *single* general-purpose task representations to select a strategy. Since it does not take the budget into account, we evaluate it only in unconstrained settings.

(3) EmbedLLM (Zhuang et al., 2024): Originally, this method uses a *single* model to predict the accuracy of each strategy and picks the one with the highest accuracy. We adapt it by adding an additional model to predict the cost. Unlike CONCUR, EmbedLLM only uses a *single* task-specific representation for computation strategies.

(4) RTR (Pan et al., 2025): This method employs a *single* model to jointly predict accuracy and cost for all strategies. In contrast to CONCUR, RTR only uses a *single* general-purpose representation for input tasks.

**Metrics.** We compare our method and the baselines based on end-to-end performance. Once computation strategies are assigned to tasks, we compute the overall accuracy and total inference FLOPs (as a proxy for efficiency and cost) for each approach. Inference FLOPs are calculated using the standard formula from Kaplan et al. (2020) and scaled down by $10^{11}$ for readability.

In the following subsections, we first evaluate our method against the baselines in non-continual unconstrained (Section 3.2) and constrained (Section 3.3) settings to primarily assess the effectiveness of our model architecture, which leverages multiple representations of both input tasks and computation strategies to improve end-to-end performance. We then examine performance in a continual setting (Section 3.4) to highlight the impact of our modular design on reducing training cost.

## 3.2 Unconstrained routing

Figure 3 shows that our method consistently outperforms the baselines in both in-distribution and out-of-distribution scenarios. Tables 2 and 3 report the maximum accuracy achieved by each method. Among the approaches that surpass the best single strategy, our method generally achieves the highest accuracy with the lowest FLOPs. This demonstrates that, compared to the best single strategy, routing enables higher accuracy at reduced computational cost. Furthermore, when compared to other routing baselines, our method delivers both superior accuracy and efficiency, highlighting its higher effectiveness as a router.

Table 2: Performance for unconstrained routing on in-distribution datasets: 2WikiMultiHop, MMLU, and GSM8k. Gray denotes methods whose *average accuracy* falls below that of the best single-strategy baseline. **Bolded** numbers indicate the best performance among the remaining methods.

| | 2WikiMultiHop | | MMLU | | GSM8k | | Average | |
|---|---|---|---|---|---|---|---|---|
| | Acc | FLOPs ↓ | Acc | FLOPs | Acc | FLOPs | Acc | FLOPs |
| Best single strategy | 57.6 | 49.63 | 73.7 | 44.45 | 91.6 | 30.82 | 74.3 | 41.63 |
| RouteLLM | 41.7 | 17.90 | 54.7 | 3.20 | 64.1 | 9.51 | 53.5 | 10.20 |
| EmbedLLM | 58.4 | 43.68 | 72.1 | 33.68 | 89.8 | 25.28 | 73.4 | 34.21 |
| RTR | 57.9 | 44.04 | 73.7 | 44.45 | 91.3 | 33.07 | 74.3 | 40.52 |
| CONCUR (Ours) | **59.5** | **38.54** | **74.4** | **40.73** | **91.6** | **30.23** | **75.2** | **36.50** |

Table 3: Performance for unconstrained routing on out-of-distribution datasets: HotpotQA, GPQA, and SVAMP.

| | HotpotQA | | GPQA | | SVAMP | | Average | |
|---|---|---|---|---|---|---|---|---|
| | Acc | FLOPs ↓ | Acc | FLOPs | Acc | FLOPs | Acc | FLOPs |
| Best single strategy | 59.8 | 39.75 | 35.0 | 75.18 | 92.2 | 23.46 | 62.3 | 46.13 |
| RouteLLM | 52.8 | 6.62 | 29.5 | 4.22 | 79.8 | 8.18 | 54.0 | 6.34 |
| EmbedLLM | 57.0 | 30.15 | 35.3 | 69.75 | 92.0 | 19.63 | 61.4 | 39.84 |
| RTR | 59.8 | 36.16 | 35.0 | 75.18 | **92.4** | 23.65 | 62.4 | 45.00 |
| CONCUR (Ours) | **60.6** | **33.62** | **35.3** | **73.31** | 92.0 | **22.75** | **62.6** | **43.23** |

Table 4: Accuracy gains of constrained routing when transitioning from local optimization (L) to global optimization (G), along with the accuracy achieved by global optimization under varying budget settings. **Bolded** and underlined numbers represent the performance of our method when it ranks as the best and second best, respectively.

| | 2WikiMultiHop | | MMLU | | GSM8k | | Average | |
|---|---|---|---|---|---|---|---|---|
| | $\Delta$(L → G) | G | $\Delta$(L → G) | G | $\Delta$(L → G) | G | $\Delta$(L → G) | G |
| *Low budget (FLOPs budget = 25)* | | | | | | | | |
| EmbedLLM | +4.4 | 52.4 | +3.2 | 70.6 | +1.5 | 89.8 | +3.0 | 70.9 |
| RTR | +6.0 | 53.9 | +4.7 | 73.7 | +2.0 | 90.4 | +4.2 | 72.7 |
| CONCUR (Ours) | +9.7 | **56.5** | +3.8 | 72.7 | +3.7 | 90.3 | +5.7 | **73.2** |
| *High budget (FLOPs budget = 40)* | | | | | | | | |
| EmbedLLM | +4.6 | 57.6 | +2.4 | 72.6 | 0.0 | 90.1 | +2.3 | 73.4 |
| RTR | +3.3 | 56.6 | +2.9 | 74.1 | +0.2 | 91.2 | +2.1 | 74.0 |
| CONCUR (Ours) | +8.1 | **59.5** | +3.2 | **74.4** | +0.8 | **91.6** | +4.0 | **75.2** |

## 3.3 CONSTRAINED ROUTING

Constrained routing means routing under budget constraints. To test generalizability, we evaluate performance under both low- and high-budget settings. Furthermore, as noted in Section 2.2, given the predicted accuracy and cost, the constrained routing problem can be addressed in two ways: (1) local optimization, which treats each task independently and allocates the budget evenly across tasks, and (2) global optimization (our approach), which distributes the total budget jointly across all tasks. We also compare the performance when using these two optimization methods.

Examining the accuracy improvement from the local optimization baseline to our global optimization method, Table 4 shows a substantial positive change in both budget settings, demonstrating the effectiveness of our global optimization approach in making better routing decisions. Moreover, when comparing the accuracy of different routing methods under global optimization, Table 4 indicates that our method achieves the highest average accuracy across both settings, highlighting the overall strength of our routing framework.

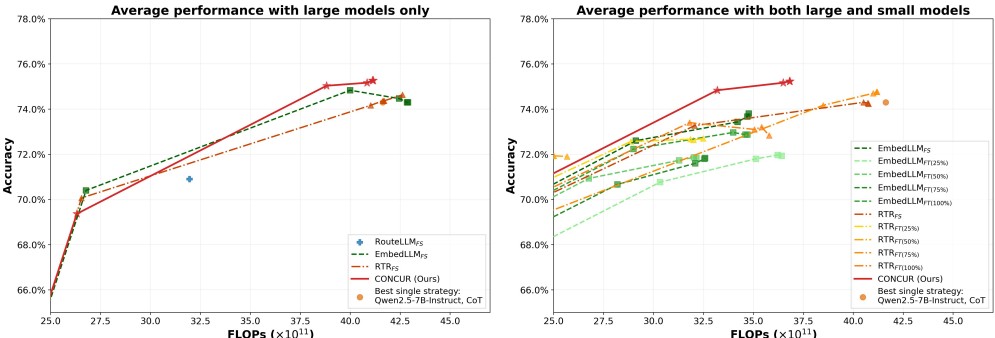

Figure 4: Performance of different methods under the continual routing with different collections of strategies. $X_{FS}$ denotes method X trained from scratch. $X_{FT(Y\%)}$ denotes method X fine-tuned from its prior version, which was trained from scratch in Setting 1, using Y% of the new data.

## 3.4 CONTINUAL ROUTING

Table 5: Performance for continual routing. Table 2 provides the definitions of the coloring and bolding scheme.

| | Relative training time | 2WikiMultiHop | | MMLU | | GSM8k | | Average | |
| --- | --- | --- | --- | --- | --- | --- | --- | --- | --- |
| | | Acc | FLOPs ↓ | Acc | FLOPs | Acc | FLOPs | Acc | FLOPs |
| *Setting 1: Large models only (Qwen2.5-7B-Instruct and Llama-3.1-8B-Instruct)* | | | | | | | | | |
| Best single strategy | – | 57.6 | 49.63 | 73.7 | 44.45 | 91.6 | 30.82 | 74.3 | 41.63 |
| RouteLLM$_{FS}$ | 0.18x | 54.0 | 48.77 | 68.7 | 17.03 | 90.0 | 30.06 | 70.9 | 31.96 |
| EmbedLLM$_{FS}$ | 1.32x | 58.9 | 49.62 | 73.3 | 44.29 | 91.2 | 33.44 | 74.5 | 42.45 |
| RTR$_{FS}$ | 3.93x | 59.4 | 51.12 | 72.1 | 42.98 | 91.6 | 30.82 | 74.4 | 41.64 |
| CONCUR (Ours) | **1.00x** | 59.3 | 50.71 | **74.5** | **41.02** | **91.7** | **30.78** | **75.2** | **40.84** |
| *Setting 2: Large and small models (Qwen2.5, Llama-3.1, and Llama-3.2 family)* | | | | | | | | | |
| Best single strategy | – | 57.6 | 49.63 | 73.7 | 44.45 | 91.6 | 30.82 | 74.3 | 41.63 |
| RouteLLM$_{FS}$ | 0.20x | 41.7 | 17.90 | 54.7 | 3.20 | 64.1 | 9.51 | 53.5 | 10.20 |
| RouteLLM$_{FT(25\%)}$ | 0.14x | 41.6 | 18.21 | 53.5 | 2.80 | 61.5 | 9.69 | 52.2 | 10.23 |
| RouteLLM$_{FT(50\%)}$ | 0.17x | 41.6 | 19.96 | 52.9 | 2.90 | 66.5 | 10.13 | 53.7 | 11.00 |
| RouteLLM$_{FT(75\%)}$ | 0.20x | 41.5 | 13.73 | 53.0 | 2.91 | 59.1 | 9.45 | 51.2 | 8.70 |
| RouteLLM$_{FT(100\%)}$ | 0.23x | 41.3 | 16.62 | 54.6 | 3.28 | 62.5 | 9.33 | 52.8 | 9.74 |
| EmbedLLM$_{FS}$ | 3.08x | 58.4 | 43.68 | 72.1 | 33.68 | 89.8 | 25.28 | 73.4 | 34.21 |
| EmbedLLM$_{FT(25\%)}$ | 1.01x | 56.9 | 40.12 | 68.9 | 34.45 | 89.6 | 30.83 | 71.8 | 35.13 |
| EmbedLLM$_{FT(50\%)}$ | 1.92x | 57.4 | 36.87 | 68.3 | 26.36 | 89.5 | 30.64 | 71.7 | 31.29 |
| EmbedLLM$_{FT(75\%)}$ | 2.82x | 55.1 | 36.19 | 69.5 | 32.69 | 90.2 | 27.40 | 71.6 | 32.09 |
| EmbedLLM$_{FT(100\%)}$ | 3.11x | 59.2 | 42.93 | 70.0 | 32.59 | 89.7 | 26.45 | 73.0 | 33.99 |
| RTR$_{FS}$ | 7.66x | 57.9 | 44.04 | 73.7 | 44.45 | 91.3 | 33.07 | 74.3 | 40.52 |
| RTR$_{FT(25\%)}$ | 1.69x | 58.5 | 39.80 | 71.3 | 42.27 | 88.2 | 13.49 | 72.7 | 31.85 |
| RTR$_{FT(50\%)}$ | 2.84x | 59.2 | 47.79 | 68.4 | 13.53 | 88.2 | 13.51 | 71.9 | 24.95 |
| RTR$_{FT(75\%)}$ | 3.89x | 58.9 | 47.81 | 73.6 | 44.38 | 91.6 | 30.82 | 74.7 | 41.01 |
| RTR$_{FT(100\%)}$ | 4.96x | 57.6 | 44.06 | 73.4 | 43.19 | 88.3 | 17.91 | 73.1 | 35.05 |
| CONCUR (Ours) | **1.00x** | **59.5** | **38.54** | **74.4** | **40.73** | **91.6** | **30.23** | **75.2** | **36.50** |

We consider the following scenarios where routers must adapt to unseen computation strategies. Initially, an organization prioritizes accuracy and selects moderate-to-large models: Qwen2.5-7B-Instruct and Llama-3.1-8B-Instruct. Later, to reduce cost and latency without sacrificing accuracy, the organization introduces smaller models: Qwen2.5-1.5B-Instruct, Qwen2.5-3B-Instruct, and Llama-3.2-3B-Instruct. We refer to the first scenario with only large models as Setting 1, and the second scenario with both large and small models as Setting 2.

Since all routing baselines (RouteLLM, EmbedLLM, and RTR) train a *single* router model using training data across *all* computation strategies, the straightforward way to incorporate unseen strategies is to retrain the model from scratch using all available data. However, this can be unnecessarily costly given that the router model has already been trained on prior data. To address this, we also

evaluated variants of these baselines where the existing router models are fine-tuned on randomly sampled 25%, 50%, 75%, and 100% of the new training data.

In addition to end-to-end performance metrics (accuracy and inference FLOPs), we explicitly measure the training time for each method as an indicator of training cost, reflecting how easily each method can adapt to unseen strategies.

Figure 4 shows that in both Setting 1 and 2, in terms of end-to-end performance, CONCUR outperforms all baselines. As shown in Table 5, in Setting 1, our method achieves the highest average accuracy with the lowest FLOPs among all baselines, demonstrating the effectiveness of our routing framework. In Setting 2, among routing baselines that outperform the best single strategy, our method again achieves the highest accuracy with the lowest FLOPs, while requiring *significantly less training time*. This highlights the advantage of our *modular* predictor architecture, which allows easy extension to unseen strategies. Importantly, the goal of the organization is to reduce FLOPs while maintaining accuracy when moving from Setting 1 to Setting 2, a target achieved only by the RTR$_{FT(75\%)}$ baseline and our method, with our approach performing substantially better.

# 4 ANALYSIS

Section 3 highlights that CONCUR outperforms existing approaches. In this section, we present a deeper analysis to better understand the sources of this improvement (Section 4.1). We also perform an ablation study to examine the effectiveness of using multiple representations for input tasks and strategies (Section 4.2). All analyses were conducted under the unconstrained routing setting discussed in Section 3.2.

## 4.1 BENEFITS OF ROUTING

Table 6: The table shows (1) the percentage of tasks routed by our framework to different strategies, (2) the performance of tasks using routed strategies by our framework compared to using the best single-strategy baseline (Qwen2.5-7B-Instruct with CoT), and (3) the distribution of task accuracy transitions from the baseline strategy to the routed strategy by our framework, where C and I indicate correct and incorrect, respectively. **Bolded** numbers indicate cases where the routed strategy by our framework outperforms the baseline strategy.

| | | Baseline strategy | | Routed strategy | | Task accuracy transitions (%) | | | |
|---|---|---|---|---|---|---|---|---|---|
| | Tasks routed (%) | Acc | FLOPs ↓ | Acc | FLOPs | C → C | I → C | I → I | C → I |
| *2WikiMultiHop* | | | | | | | | | |
| Qwen7B-CoT (Baseline) | 33.6% | 82.4 | 47.7 | – | – | – | – | – | – |
| Qwen2B-vanilla | 25.3% | 22.9 | 56.9 | **27.3** | **7.8** | 16.2% | 11.1% | 66.0% | 6.7% |
| Llama8B-CoT | 21.0% | 54.8 | 48.6 | **56.7** | 70.5 | 48.1% | 8.6% | 36.7% | 6.7% |
| Qwen7B-vanilla | 17.1% | 67.3 | 43.5 | **70.2** | **29.9** | 59.1% | 11.1% | 21.6% | 8.2% |
| Others | 3.0% | 36.7 | 51.8 | 33.3 | **21.0** | 23.3% | 10.0% | 53.3% | 13.3% |
| *MMLU* | | | | | | | | | |
| Qwen7B-CoT (Baseline) | 83.6% | 74.2 | 44.4 | – | – | – | – | – | – |
| Qwen7B-vanilla | 12.7% | 69.3 | 44.5 | **78.0** | **14.4** | 64.6% | 13.4% | 17.3% | 4.7% |
| Others | 3.7% | 78.4 | 45.3 | 67.6 | 48.1 | 62.2% | 5.4% | 16.2% | 16.2% |
| *GSM8k* | | | | | | | | | |
| Qwen7B-CoT (Baseline) | 96.0% | 91.6 | 30.9 | – | – | – | – | – | – |
| Others | 4.0% | 92.5 | 27.9 | 92.5 | **13.2** | 87.5% | 5.0% | 2.5% | 5.0% |

Section 3 demonstrates that CONCUR outperforms the best single-strategy baseline (Qwen2.5-7B-Instruct with CoT decoding) in both accuracy and efficiency. In this section, we take a closer look to understand the source of gain. Specifically, we analyze the routing decisions made by our framework in comparison to the best single-strategy baseline, which routes all tasks to one fixed computation strategy. We also analyze how task accuracy varies between the baseline and our framework.

As shown in Table 6, our framework assigns most GSM8k tasks to Qwen7B-CoT, so performance remains similar compared to the best single-strategy baseline. However, for 2WikiMultiHop and MMLU, our framework routes a substantial portion of tasks to more cost-efficient strategies (smaller

models and/ or simpler decoding methods). These alternatives often achieve higher accuracy while significantly reducing FLOPs, explaining the performance gains of our framework.

Additionally, Table 6 shows how tasks are distributed according to their accuracy changes when moving from the best single-strategy baseline to our framework, indicating whether they stay correct/incorrect or transition between the two. Most tasks follow one of two patterns: (1) they keep their original correctness but are routed to cheaper strategies, significantly reducing computation cost and improving efficiency; or (2) they switch from previously incorrect to correct answers, leading to gains in accuracy. A small fraction of tasks change from correct to incorrect, but these losses are minor compared to the overall improvements. This again illustrates the source of our gains.

## 4.2 ABLATION STUDY ON REPRESENTATIONS

As discussed in Section 2.1, our framework considers two types of representations for input tasks and strategies: a general-purpose representation and a task-specific representation. To verify the contribution of each, we evaluate our framework using only the general-purpose representation, only the task-specific representation, and the combination of both. As shown in Table 7, each representation provides valuable information, while their combination consistently delivers the best accuracy. This demonstrates that the two representations offer complementary signals and that leveraging both is crucial for realizing the full effectiveness of our framework.

Table 7: Performance of CONCUR in unconstrained routing when using different representations for both input tasks and strategies. **Bolded** and underlined numbers represent the performance of our framework using both representations when it ranks as the best and second best, respectively.

| | 2WikiMultiHop | | MMLU | | GSM8k | | Average | |
|---|---|---|---|---|---|---|---|---|
| | Acc | FLOPs ↓ | Acc | FLOPs | Acc | FLOPs | Acc | FLOPs |
| General-purpose only | 57.9 | 43.82 | 72.5 | 25.71 | 89.5 | 37.05 | 73.3 | 35.53 |
| Task-specific only | 58.7 | 45.69 | 68.5 | 29.01 | 89.8 | 37.25 | 72.3 | 37.32 |
| Both (CONCUR) | **59.5** | **38.54** | **74.4** | 40.73 | **91.6** | **30.23** | **75.2** | 36.50 |

## 5 CONCLUSION

This work introduces CONCUR, a framework for continual constrained and unconstrained routing. Central to CONCUR are modular predictors that leverage both general-purpose and task-specific representations to estimate a strategy's accuracy and cost on a given task, enabling optimization-based routing and straightforward extension to unseen strategies. Extensive experiments on a diverse set of in-distribution and out-of-distribution tasks show that CONCUR outperforms the best single strategy and existing strong routing methods in both continual and non-continual settings.

## ACKNOWLEDGMENTS

We gratefully acknowledge support from the DARPA ASKEM Award HR00112220042, the ARPA-H Biomedical Data Fabric project, Liberty Mutual, Jane Street, the MIT MGAIC Consortium, the MIT Everest Projects, the MIT-IBM Watson AI Lab, ONR-LtRD (N00014-23-1-2364), and the Google PhD Fellowship in collaboration with MIT. We also thank Amazon, Google, and Intel for their contributions through the MIT Data Systems and AI Lab (DSAIL).

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

# A  DESCRIPTIONS FOR COMPUTATION STRATEGY

Table 8 lists the description for decoding strategies and Table 9 lists the descriptions for models.

Table 8: Descriptions of decoding strategies.

| Strategy ID | Description |
|---|---|
| Vanilla | Vanilla prompting retains the original question content without adding any additional prompt information. |
| CoT | Chain-of-Thought (CoT) prompting guides the model to articulate a step-by-step reasoning process before providing the final answer. This results in longer responses and slower inference, but delivers superior performance on complex reasoning tasks. |

Table 9: Descriptions of models.

| Model ID | Description |
|---|---|
| Qwen2.5-1.5B-Instruct | Qwen2.5-1.5B-Instruct is an ultra-lightweight 1.5 B parameter model designed for minimal-resource environments. It is best suited for simple prompts, basic classification, and short text completion, but struggles with nuanced understanding or advanced reasoning tasks. |
| Qwen2.5-3B-Instruct | Qwen2.5-3B-Instruct is a lightweight 3 B parameter model with fast inference and low resource usage. It is suitable for simple tasks such as basic question answering and short-form text generation, but is limited in handling complex reasoning or multi-step tasks. |
| Qwen2.5-7B-Instruct | Qwen2.5-7B-Instruct is a mid-small 7 B parameter model that balances speed and performance. It can handle multi-turn dialogue, basic code and math tasks, and offers improved language understanding over smaller models while maintaining efficient inference. |
| Llama-3.2-3B-Instruct | Llama-3.2-3B-Instruct is a compact 3 B parameter model optimised for efficient inference in constrained environments. It handles basic instruction following, simple question answering, and short text generation reliably, but lacks the depth for nuanced reasoning or complex task execution. |
| Llama-3.1-8B-Instruct | Llama-3.1-8B-Instruct is a moderately-sized 8 B parameter model that offers a strong balance between performance and resource usage. It supports multi-turn dialogue, intermediate reasoning, and modest code or math capabilities, though it may still struggle with deeply intricate or highly technical prompts. |

## B    DYNAMIC PROGRAMMING FORMULATION FOR SOLVING CONSTRAINED OPTIMIZATION

To address the constrained optimization in Equation (2), we formulate dynamic programming (DP)-based solutions. We define $DP[i][b]$ as the maximum achievable total accuracy when routing the first $i$ tasks, subject to a total cost not exceeding $b$.

Then, we initialize the DP problem as follows

$$DP[0][b] = \begin{cases} 0 & \text{if } b = 0 \\ -\infty & \text{otherwise} \end{cases} \quad \forall b \in [0, nB] \tag{3}$$

We define the recurrence relation as follows: for all $b \in [B_i^{\min}, \min\left(B_i^{\max}, nB\right)]$

$$DP[i][b] = \max_{j, b \geq c_{ij}} DP[i-1][b - c_{ij}] + a_{ij} \tag{4}$$

where $B_i^{\min} = \sum_{k=0}^{i} \min_j c_{kj}$ and $B_i^{\max} = \sum_{k=0}^{i} \max_j c_{kj}$. $B_i^{\min}$ and $B_i^{\max}$ denote the minimum and maximum total cost required to assign exactly one method to each of the first $i$ tasks. We constrain $b$ within these bounds to avoid unnecessary computations. This recurrence reflects the process of updating the maximum cumulative accuracy by considering all computation strategies for task $t_i$ and choosing the one that achieves the highest accuracy without exceeding the budget. Since the budget $nB$ and the cost $c_{ij}$ can be floating-point numbers, we round them to integers to enable integer-based indexing in the DP array.

Then, the maximum accuracy attainable within the budget $nB$ is $\max_{b \leq nB} DP[n][b]$. We apply backtracking to recover the strategy chosen for each task.

## C    PROMPTS

Tables 10 to 12 show the prompts used for all task types (multi-hop QA, general reasoning, and math problems) with different decoding strategies.

Table 10: Prompts for multi-hop QA (blue refers to the vanilla prompt and yellow refers to the CoT prompt).

---

**2WikiMultiHop and HotpotQA (vanilla & CoT)**

**System:** You are an expert at question answering.

**User:**

You are provided with a user question, and information that might be relevant to the user question.Your task is to *only* output a short answer within `<ans></ans>`.

You are provided with a user question, and information that might be relevant to the user question.Please *reason step by step* before providing the short answer; put your final answer within `<ans></ans>`.

*Document title:* **Mistress (1992 film)**
*Document content:* Robert De Niro is the producer of the film *Mistress*.
*Document title:* **The Godfather Part II**
*Document content:* Robert De Niro played the role of Vito Corleone in *The Godfather Part II*.
Here is the user question:
  *In* The Godfather Part II*, who did the producer of* Mistress *play?*

---

Table 11: Prompts for general reasoning.

---

**MMLU and GPQA (vanilla & CoT)**

**System:** You are an expert at question answering.

**User:**

You are provided with a multi-choice question. Your task is to *only* output an answer (the letter corresponding to the answer choice placed inside parentheses) within `<ans></ans>` (e.g. `<ans>(A)</ans>`).

You are provided with a multi-choice question. Please *reason step by step* before providing the final answer, and put your final answer (the letter corresponding to the answer choice placed inside parentheses) within `<ans></ans>`.

Here is the user question:
Which of the following is a second messenger that stimulates release of calcium ions into the cytoplasm?

Here are the multiple-choice answers:
(A) Prostaglandins
(B) Inositol triphosphate
(C) Cyclic AMP
(D) Calmodulin

---

Table 12: Prompts for math problems.

---

**GSM8K and SVAMP (vanilla & CoT)**

**System:** You are an expert at solving math questions.

**User:**

You are provided with a math question. Your task is to *only* output a numerical answer within `<ans></ans>`.

You are provided with a math question. Please *reason step by step* before providing a numerical answer; put your final answer within `<ans></ans>`.

Here is the user question:
Tommy is fundraising for his charity by selling brownies for $3 a slice and cheesecakes for $4 a slice. If Tommy sells 43 brownies and 23 slices of cheesecake, how much money does Tommy raise?

---

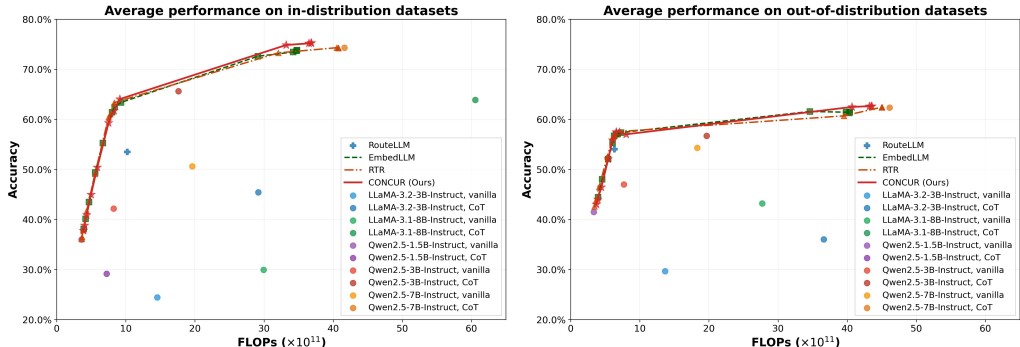

Figure 5: Performance of all methods for unconstrained routing on both in- and out-of-distribution datasets across various values of $w$ defined in Section 2.2, illustrating the trade-off between accuracy and cost.

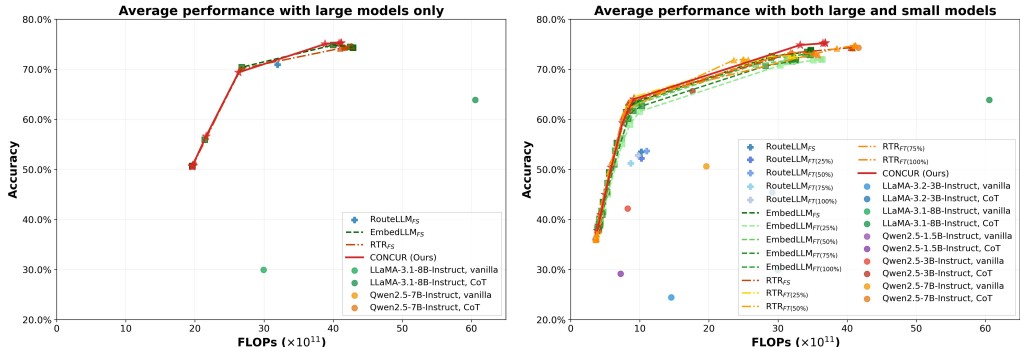

Figure 6: Performance of all methods under the continual setting with different collections of strategies. $X_{FS}$ denotes method X trained from scratch. $X_{FT(Y\%)}$ denotes method X fine-tuned from its prior version, which was trained from scratch in Setting 1, using Y% of the new data.

## D    IMPLEMENTATION DETAILS

We used the off-the-shelf ALL-MPNET-BASE-V2[2] model as the frozen encoder outlined in Section 2.1, following the approach in Pan et al. (2025); Zhuang et al. (2024), which generates representations of size $k = 768$. Training was conducted on an A100 GPU cluster for up to 100 epochs, using the Adam optimizer with a batch size of 32 and an initial learning rate of $1 \times 10^{-3}$.

## E    FULL DIAGRAMS

Figures 5 and 6 present the full versions of Figures 3 and 4, respectively, including all methods.

## F    THE USE OF LARGE LANGUAGE MODELS (LLMS)

LLM was used only to aid writing quality (proofreading and polishing grammar). No ideas, claims, methods, results, or references are generated by LLMs. All content decisions and revisions are made by the authors.

---

[2]https://huggingface.co/sentence-transformers/all-mpnet-base-v2

