# OpenReview forum: "CONCUR: A Framework for Continual Constrained and Unconstrained Routing"
_ICLR.cc/2026/Conference — ICLR 2026 Poster_

### Official Review · Reviewer_DE3V · 2025-10-16

**Soundness:** 3
**Presentation:** 3
**Contribution:** 3
**Rating:** 4
**Confidence:** 3

**Summary:**

This paper introduces CONCUR, a modular framework for continual task routing among diverse computation strategies, addressing the challenge of efficiently mapping AI tasks to optimal computational configurations under both constrained (budget-aware) and unconstrained conditions. Unlike traditional monolithic routing approaches that require retraining a single model whenever new strategies are added, CONCUR proposes a modular design in which each computation strategy has its own predictor model estimating its accuracy and cost on a given task. These predictors rely on both general-purpose and task-specific representations, improving the granularity and flexibility of routing decisions.

**Strengths:**

1. The modular predictor design allows new strategies to be added by training only their respective predictors, greatly reducing retraining cost and time compared to monolithic routers.
2. Combines general-purpose embeddings (from pretrained encoders) with learnable task- and strategy-specific representations, demonstrating superior expressiveness over single-representation baselines.
3. Comprehensive evaluation and analysis.

**Weaknesses:**

1. While modularity is beneficial, training two predictors (accuracy + cost) per strategy may become cumbersome when the number of strategies grows very large.
2. The paper lacks a deeper theoretical justification for why modular predictors generalize better than joint models and the optimization formulations are straightforward but not novel in theory.
3. The predictors act as black boxes; it remains unclear how they capture task–strategy interactions or why they perform well in OOD tasks.

**Questions:**

See weakness

---

> ### Author Response · Authors · 2025-12-03
> **Author response 1 (A1)**
>
> We sincerely appreciate the reviewer’s thoughtful feedback. We appreciate that the advantages of our modular predictor design and the benefits gained from combining general-purpose and task/strategy-specific representations were clear. We also appreciate the reviewer’s recognition of the breadth of our evaluation. We hope the clarifications and additional context provided below reinforce the soundness of our method and help address the concerns raised.
>
> > W1: While modularity is beneficial, training two predictors (accuracy + cost) per strategy may become cumbersome when the number of strategies grows very large.
>
> We agree that a very large number of strategies increases the number of predictors. However, training modular predictors typically scales better than training monolithic routers proposed by existing work because of the following reasons:
> - **Efficient incremental updates**: Monolithic routers typically require full retraining whenever a strategy changes or a new strategy is added, creating substantial repeated overhead. In our modular design, only the two small predictors (accuracy and cost) for the new or modified strategy need retraining, making updates far more efficient.
> - **Fully parallelizable training**: Each predictor in the modular design is trained independently on its own strategy’s data, enabling fully parallel training across machines or accelerators. This avoids the sequential, memory-intensive training required by a monolithic router and ensures that incremental updates load only the relevant strategy data, significantly reducing compute and memory usage.
>
> The above benefits are reflected empirically: as shown in Table 5 in the paper, our approach achieves significantly lower training time compared to monolithic routers, particularly in settings with evolving or expanding strategies.

---

> ### Author Response · Authors · 2025-12-03
> **Author response 2 (A2)**
>
> > W2: The paper lacks a deeper theoretical justification for why modular predictors generalize better than joint models
>
> > W3: why they perform well in OOD tasks
>
> We thank the reviewer for the request for deeper theoretical justification. The empirical superiority of modular predictors over joint models (e.g., RouteLLM, RTR) can be understood through recent theory (Boopathy et al., 2024 [1]). Boopathy et al. define a task as modularly structured if its target function can be decomposed into independent sub-functions. In our setting, the goal of predicting accuracy and cost for all strategies on a task can naturally be decomposed into independent sub-goals, predicting accuracy and cost for each strategy separately, so our problem satisfies this notion of modularity.
>
> A key theoretical insight discovered by Boopathy et al. is that as task dimensionality increases, training modular models becomes much easier than training joint models, leading to better generalization both in-distribution and out-of-distribution. For our framework, a joint model must consider all strategies simultaneously, resulting in a high task dimension, whereas each modular predictor only handles a single strategy, greatly reducing task dimensionality. This provides a theoretical explanation for why our modular design generalizes more effectively and performs well on OOD tasks. We will add a more thorough discussion in our revised paper.
>
> [1] Boopathy, Akhilan, et al. "Breaking neural network scaling laws with modularity." arXiv preprint arXiv:2409.05780 (2024).
>
> > W2: The optimization formulations are straightforward but not novel in theory.
>
> While the optimization formulations themselves are straightforward, the novelty of our work lies in addressing key limitations of model architectures proposed by existing routing frameworks: (1) limited scalability caused by monolithic models, and (2) restricted parametrizations that reduce effectiveness. To address these limitations, our key contribution is a novel modular architecture that utilizes multiple representations of input tasks and strategies, which, as demonstrated in our experiments, consistently outperforms previous routing systems across a wide range of tasks.
>
> We adopt the predict-then-optimize approach rather than end-to-end learning proposed in prior work (e.g., RouteLLM) because it (a) allows users to explicitly incorporate constraints, and (b) achieves substantially better performance than end-to-end approaches as seen in experiments.

---

> ### Author Response · Authors · 2025-12-03
> **Author response 3 (A3)**
>
> > W3: The predictors act as black boxes; it remains unclear how they capture task–strategy interactions
>
> The predictors combine both general-purpose representations and task-specific representations of the input tasks and strategies. The general-purpose representations encode broad characteristics shared across tasks and strategies, while the task-specific representations encode fine-grained, strategy-dependent features.
>
> The MLP in each predictor takes these representations as input and outputs the estimated accuracy and cost, acting as a flexible function approximator that learns how the combined features jointly influence the predictions. In other words, the MLP models complex, nonlinear interactions between task and strategy characteristics, allowing the framework to accurately predict outcomes for diverse task-strategy pairs.

---

### Official Review · Reviewer_dMBg · 2025-10-29

**Soundness:** 3
**Presentation:** 2
**Contribution:** 3
**Rating:** 4
**Confidence:** 3

**Summary:**

This paper introduces a unified framework for routing among multiple LLMs and decoding strategies under both continual and non-continual settings. The main idea is to train modular predictors that estimate, for each model–decoding pair, its expected accuracy and computational cost on a given task. These predictors combine general text embeddings (from an off-the-shelf embedding model) with task-specific representations to capture both semantic and domain-specific information. Routing decisions are then formulated as an optimization problem that can be solved efficiently. Experimental results show that the proposed framework outperforms existing baselines in terms of both accuracy and efficiency.

**Strengths:**

- The paper addresses an important and practical problem. Effectively routing among multiple LLMs and decoding strategies under varying cost and performance constraints has become an increasingly relevant and widely discussed topic. This topic is relevant to the growing need for efficient deployment of large reasoning models, especially in settings where computational budgets and task distributions evolve over time. The formulation of both continual and non-continual routing scenarios adds further practical value to the work.

- The proposed framework demonstrates consistent improvements over existing routing baselines across several reasoning and question-answering benchmarks. The experimental results suggest that the modular predictor design and optimization-based decision process lead to better accuracy–efficiency trade-offs, validating the effectiveness of the proposed approach in diverse routing settings.

**Weaknesses:**

- Although the paper repeatedly discusses routing in both continual and non-continual settings, the problem itself is never formally defined. It remains unclear what exactly constitutes a routing input and output, and how the optimization objective is formulated in mathematical terms. The framework is presented primarily at a conceptual level, which makes it difficult to precisely understand the problem scope and to reproduce the approach in other contexts. As a result, the paper may not be self-contained for readers who are not already familiar with prior routing works such as RouteLLM or RTR, which substantially limits accessibility and clarity.

- The overall framework follows a structure that is already common in related areas such as task planning and model selection. The paper mainly adapts existing ideas to the LLM routing setting, without introducing a fundamentally new algorithmic or theoretical insight.

- The paper lacks a systematic ablation study quantifying the contribution of each component in the proposed framework. Is there any ablation study or experimental analysis addressing this? For example, how would the performance change if only the general-purpose representation or the task-specific representation were used?

**Questions:**

See the weakness above.

---

> ### Author Response · Authors · 2025-12-03
> **Author response 1 (A1)**
>
> We thank the reviewer for the thoughtful feedback. We appreciate the reviewer highlighting the practical importance of efficient routing across LLMs and decoding strategies, as well as the relevance of addressing both continual and non-continual deployment scenarios. We also appreciate that the reviewer recognized the consistent empirical gains achieved by our modular predictor framework and optimization-based routing procedure. We hope the clarifications and additional context provided below reinforce the soundness of our method and help address the concerns raised.
>
> > W1: Although the paper repeatedly discusses routing in both continual and non-continual settings, the problem itself is never formally defined and how the optimization objective is formulated in mathematical terms.
>
> Our work considers two routing problems: constrained routing and unconstrained routing. The continual/non-continual settings describe the temporal context in which these routing tasks are performed. Specifically, in the continual setting, the framework is trained on an initial set of strategies (a strategy refers to a specific language model and a decoding method) at time X and must later incorporate additional strategies that become available at a later time Y. In contrast, in the non-continual setting, the set of strategies remains fixed over time.
>
> The formal definitions of the two routing problems are provided in Section 2.2. For unconstrained routing, the optimization objective is to maximize accuracy $a_{ij}$ and minimize computational cost $c_{ij}$, expressed in line 210 as $\max a_{ij} \min c_{ij}$, where $i$ represents the task and $j$ represents the strategy. For constrained routing, the objective is defined in line 218 as $\max_j a_{ij}$  such that $c_{ij} ≤ B$ where $B$ is the cost budget.
>
> > W1: It remains unclear what exactly constitutes a routing input and output.
>
> As shown in Figure 1 and implied by the optimization problems above, the input consists of (i) a downstream task and (ii) a candidate set of computational strategies (a strategy refers to a specific language model and a decoding method). The output is the selected strategy from the input set that best satisfies the corresponding routing objective (either the unconstrained or constrained formulation). We will make it clearer in the revised version of our paper.
>
> > W1: The framework is presented primarily at a conceptual level, which makes it difficult to precisely understand the problem scope and to reproduce the approach in other contexts. As a result, the paper may not be self-contained for readers who are not already familiar with prior routing works such as RouteLLM or RTR, which substantially limits accessibility and clarity.
>
> Thank you for your suggestions. We will improve the presentation of the paper.

---

> ### Author Response · Authors · 2025-12-03
> **Author response 2 (A2)**
>
> > W2: The overall framework follows a structure that is already common in related areas such as task planning and model selection. The paper mainly adapts existing ideas to the LLM routing setting, without introducing a fundamentally new algorithmic or theoretical insight.
>
> We appreciate the reviewer’s comment that LLM routing is conceptually related to areas such as task planning and model selection, and we agree that routing can support these areas. However, the core contribution of this work is to advance LLM routing frameworks themselves. As discussed in Section 1, existing LLM routing approaches face two key limitations that our method directly addresses:
> * **Limited scalability due to monolithic routing models**: Prior routing frameworks typically use a single model trained jointly across all strategies, which makes incorporating new strategies or models difficult. This motivates our modular architecture, which trains one predictor for each strategy. As shown in Section 3.4, this modularity substantially reduces training time and improves extensibility.
> * **Restricted parametrizations that limit effectiveness**: Existing routers often rely on a single representation or limited parameterization, which can lead to suboptimal strategy predictions. This motivates our use of multiple representations to improve the quality of strategy predictions. Sections 3.2 and 3.3 demonstrate that this design improves downstream performance.
>
> To summarize, our key contribution is a novel modular architecture that utilizes multiple representations of input tasks and strategies, which, as demonstrated in our experiments, consistently outperforms previous routing systems across a wide range of tasks.

---

> ### Author Response · Authors · 2025-12-03
> **Author response 3 (A3)**
>
> > W3: The paper lacks a systematic ablation study quantifying the contribution of each component in the proposed framework. Is there any ablation study or experimental analysis addressing this? For example, how would the performance change if only the general-purpose representation or the task-specific representation were used?
>
> The table below reports the performance when using only the general-purpose representation, only the task-specific representation, and both together. As shown, each component contributes useful information, but the combination consistently yields the strongest performance. This confirms that the two representations capture complementary signals and that using both is essential for achieving the full effectiveness of our framework. We will add this table to the revised paper.
>
> |                 | 2WikiMultiHop Accuracy | 2WikiMultiHop FLOPs | MMLU Accuracy | MMLU FLOPs | GSM8k Accuracy | GSM8k FLOPs | Average Accuracy | Average FLOPs |
> |-------------------------|-----------|-------------|----------|------------|-----------|-------------|---------|-----------|
> | General-purpose + Task-specific | 59.5 | 38.54 | 74.4 | 40.73 | 91.6 | 30.23 | 75.2 | 36.5 |
> | General-purpose only    | 57.9      | 43.82       | 72.5     | 25.71      | 89.5      | 37.05       | 73.3    | 35.53     |
> | Task-specific only      | 58.7      | 45.69       | 68.5     | 29.01      | 89.8      | 37.25       | 72.3    | 37.32     |

---

### Official Review · Reviewer_NPzC · 2025-10-31

**Soundness:** 3
**Presentation:** 3
**Contribution:** 3
**Rating:** 6
**Confidence:** 3

**Summary:**

This paper proposes CONCUR, a continual routing framework designed to address limitations in existing AI task routing systems, supporting both constrained (with cost budget) and unconstrained (no budget) routing scenarios. CONCUR outperforms baselines in both unconstrained and constrained routing.

**Strengths:**

1.	The modular predictor design directly solves the retraining overhead issue of existing methods. Integrating new strategies with minimal cost aligns with real-world needs (e.g., frequent updates of language models), making it highly scalable.

2.	By combining general-purpose and task-specific representations, CONCUR captures both universal patterns (via general embeddings) and task/strategy-specific nuances (via learnable representations), which is empirically shown to improve routing decisions (e.g., higher accuracy on out-of-distribution tasks).

3.	Supporting both constrained and unconstrained scenarios broadens its applicability—unconstrained for use cases prioritizing performance balance, and constrained for resource-limited environments (e.g., edge devices with tight FLOP budgets).

4.	The study uses diverse datasets (covering different task types and distribution shifts), standardized metrics (accuracy + FLOPs), and comparisons to strong baselines, ensuring the validity and generalizability of results.

**Weaknesses:**

1.	The general-purpose representation relies on a frozen pre-trained embedding model (ALL-MPNET-BASE-v2). The paper does not evaluate how changes to this model (e.g., using a smaller/larger embedding model) affect predictor performance, raising concerns about its robustness to embedding choice.
2.	The unconstrained routing uses a weight to balance accuracy and cost, but the paper does not analyze how different weight values (e.g., weight=0.1 for cost prioritization vs. weight=0.9 for accuracy prioritization) impact real-world utility, nor does it provide guidance on choosing this weight for specific use cases.

**Questions:**

1.	How does replacing the ALL-MPNET-BASE-v2 model with other embedding models affect the accuracy and cost prediction of CONCUR? Is there a minimum embedding quality required for the framework to perform well?

2.	When integrating a completely new strategy with no historical task data (cold start), how does CONCUR perform? Do you have any initialization methods (e.g., transferring knowledge from similar strategies) to improve the cold-start prediction accuracy of new predictors?

---

> ### Author Response · Authors · 2025-12-03
> **Author response 1 (A1)**
>
> We thank the reviewer for the detailed and positive feedback. We appreciate that the advantages of our modular predictor design, the effectiveness of combining general-purpose and task-specific representations, and the practicality of supporting both constrained and unconstrained routing scenarios were clearly conveyed. We hope the clarifications and additional context provided below reinforce the soundness of our method and help address the concerns raised.
>
> > W1: The general-purpose representation relies on a frozen pre-trained embedding model (ALL-MPNET-BASE-v2). The paper does not evaluate how changes to this model (e.g., using a smaller/larger embedding model) affect predictor performance, raising concerns about its robustness to embedding choice.
>
> > Q1: How does replacing the ALL-MPNET-BASE-v2 model with other embedding models affect the accuracy and cost prediction of CONCUR? Is there a minimum embedding quality required for the framework to perform well?
>
> We originally selected `ALL-MPNET-BASE-v2` because it was the embedding model used in prior frameworks: EmbedLLM and RTR. To assess robustness to both model size and embedding family, we have run an additional experiment where we evaluated two widely used alternatives: `BGE-small-en-v1.5` (a smaller 33.4M-parameter model) and `multilingual-e5-base` (a larger 0.3B-parameter model). For reference, the original `MPNet` model has approximately 0.1B parameters.
>
> The table below shows the highest accuracy attained by our method, along with the corresponding FLOPs, for each embedding model. It shows that using either of these models in place of `MPNet` yields comparable accuracy and cost-prediction performance. Importantly, these models span a broad range of embedding quality on the MTEB leaderboard (https://huggingface.co/spaces/mteb/leaderboard), from scores of 42.62 to 57.02 and rankings from 39th to 106th, yet the performance of our framework remains stable across this spectrum. These results suggest that our framework is generally robust to embedders of different sizes, architectures, and quality levels.
>
> |                          | 2WikiMultiHop Accuracy | 2WikiMultiHop FLOPs | MMLU Accuracy | MMLU FLOPs | GSM8k Accuracy | GSM8k FLOPs | Average Accuracy | Average FLOPs |
> |--------------------------------|-----------|-------------|----------|------------|-----------|-------------|---------|-----------|
> | `all-mpnet-base-v2`                 | 59.5      | 38.54       | 74.4     | 40.73      | 91.6      | 30.23       | 75.2    | 36.5      |
> | `bge-small-en-v1.5`     | 58.7      | 43.99       | 73.7     | 32.63      | 91.6      | 30.82       | 74.7    | 35.8      |
> | `multilingual-e5-base`  | 58.2      | 45.90       | 73.5     | 32.71      | 91.2      | 30.41       | 74.3    | 36.3      |

---

> ### Author Response · Authors · 2025-12-03
> **Author response 2 (A2)**
>
> > W2: The unconstrained routing uses a weight to balance accuracy and cost, but the paper does not analyze how different weight values (e.g., weight=0.1 for cost prioritization vs. weight=0.9 for accuracy prioritization) impact real-world utility, nor does it provide guidance on choosing this weight for specific use cases.
>
> In the unconstrained setting, the weight $w$ explicitly encodes the trade-off between accuracy and computational cost, and Figures 3-6 in the paper illustrate how varying $w$ produces different points along the Pareto frontier. Small values of $w$ prioritize cost, resulting in low-FLOP/low-accuracy strategies, while larger values emphasize accuracy, selecting higher-cost strategies. These Pareto curves directly reflect the practical outcomes of different weight choices, allowing practitioners to select $w$ according to their deployment needs. For instance, resource-constrained or latency-sensitive applications may choose a smaller $w$ to save computation, while high-stakes tasks may use a larger $w$ to prioritize accuracy. In practice, users can select $w$ based on FLOP/latency budgets, minimum acceptable accuracy, or the "elbow" of the Pareto curve to achieve a balanced trade-off. For instance, examining the Pareto curve in the left panel of Figure 3, a user aiming to maximize accuracy while keeping FLOPs no more than 35.0 would select the weight $w$ that yields an accuracy of approximately 75.0%.

---

> ### Author Response · Authors · 2025-12-03
> **Author response 3 (A3)**
>
> > Q2: When integrating a completely new strategy with no historical task data (cold start), how does CONCUR perform? Do you have any initialization methods (e.g., transferring knowledge from similar strategies) to improve the cold-start prediction accuracy of new predictors?
>
> We appreciate the reviewer’s thoughtful question. Our paper focuses on improving existing training-based routing frameworks so they can operate effectively in a continual setting. Designing a routing framework that remains effective even in training-free or cold-start scenarios is indeed an interesting and valuable direction. We think exploring mechanisms such as training-free initialization or knowledge transfer from similar strategies is an important area for future extension.

---

### Official Review · Reviewer_d5bV · 2025-10-31

**Soundness:** 3
**Presentation:** 3
**Contribution:** 3
**Rating:** 6
**Confidence:** 3

**Summary:**

This paper introduces a routing method to choose the best model, maximizing efficiency and accuracy, for a continual learning setup, where the task based routing predictor can be updated to include new model variations (strategies) on the fly.

**Strengths:**

1. The paper addresses a timely problem of efficiently routing between a growing variety of models to solve a task with maximum accuracy and efficiency (measured in FLOPs).
2. The proposed method of learning a model specific predictor without retraining the strategy prediction backbone makes routing to new models seamless as also verified by the experiments in the paper.

**Weaknesses:**

1. The authors use FLOPs to measure and learn the efficiency while choosing between models. Are the FLOPs measured by counting the number of multiply-add operations? If that is the case, the proposed method will not capture the efficiency achieved by quantized models, which can be much more efficient and still achieve similar accuracies (most models are now released with lightweight quantized version).
2. It is unclear to me, how the base strategy predictor performs when the number of input strategies increase, i.e, what is the expressivity of this predictor and does it overfit to initial learnt strategies?
3. Maybe this is a more high level question, but how would routing work in the case where the questions (tasks) are open ended and there does not exist a right answer to measure accuracy?

**Questions:**

See above.

---

> ### Author Response · Authors · 2025-12-03
> **Author response 1 (A1)**
>
> We thank the reviewer for the thoughtful and encouraging feedback. We appreciate the reviewer’s recognition of the relevance of the problem and the benefits of our modular routing design. We hope the clarifications and additional context provided below reinforce the soundness of our method and help address the concerns raised.
>
> > W1: The authors use FLOPs to measure and learn the efficiency while choosing between models. Are the FLOPs measured by counting the number of multiply-add operations? If that is the case, the proposed method will not capture the efficiency achieved by quantized models, which can be much more efficient and still achieve similar accuracies (most models are now released with lightweight quantized version).
>
> In the paper, we measure efficiency using FLOPs, computed in terms of multiply-add operations. We chose FLOPs because it is a model-agnostic metric for comparing model architectures of different sizes. Since our experiments do not include quantized models, FLOPs offer a fair basis for comparison in this setting.
>
> However, our solution is not restricted to FLOPs. FLOPs can be substituted or combined with other numerical efficiency metrics, such as throughput, latency, or energy consumption, and our current method can predict these. Thus, different metrics can be used depending on the scenario, including cases involving quantized models.

---

> ### Author Response · Authors · 2025-12-03
> **Author response 2 (A2)**
>
> > W2: It is unclear to me, how the base strategy predictor performs when the number of input strategies increase, i.e, what is the expressivity of this predictor and does it overfit to initial learnt strategies?
>
> Our framework adopts a modular architecture in which each candidate strategy is equipped with its own predictor that estimates accuracy and efficiency for a given task. Because each predictor only models a single strategy, its expressivity does not diminish as the number of strategies increases; adding more strategies simply adds more independent predictors rather than increasing the complexity of any existing predictor. This design also reduces the potential for overfitting because each predictor learns the task-performance relationship for its own strategy without being biased by others.
>
> Empirically, our experiments already demonstrate that the method generalizes well: (1) on a fixed set of strategies across diverse tasks (Sections 3.2 and 3.3) both in-distribution and out-of-distribution, and (2) when increasing the number of strategies (Section 3.4). In both settings, our approach consistently outperforms the baselines.

---

> ### Author Response · Authors · 2025-12-03
> **Author response 3 (A3)**
>
> > W3: Maybe this is a more high level question, but how would routing work in the case where the questions (tasks) are open ended and there does not exist a right answer to measure accuracy?
>
> For open-ended tasks where there is no single “correct” answer and accuracy is not well-defined, response quality can still be evaluated using various methods. Our routing framework can incorporate these quality measures (whether binary or continuous) and select models that best balance both the quality and efficiency. For example, LLM-as-a-judge is a common approach in which an LLM scores responses according to predefined criteria, enabling relatively consistent quality assessments even for open-ended queries.

---

### Author Response · Authors · 2025-12-03
**Rebuttal summary**

We thank the reviewers for their thoughtful and constructive feedback.

Existing LLM routing frameworks face two key limitations: (1) limited scalability due to monolithic models, which significantly increases training cost, particularly in continual settings when new strategies (language model-decoding method pairs) are introduced, and (2) restricted parametrizations that reduce routing effectiveness.
As our contribution, we propose a novel modular routing architecture that trains one predictor for each strategy and leverages multiple representations of input tasks and strategies. Across a diverse set of datasets (including multi-hop QA, general reasoning, and math problems) and under both in-distribution and out-of-distribution settings, as well as in continual and non-continual scenarios, our experiments show that the proposed framework consistently outperforms prior methods in terms of both training efficiency and routing quality.

All reviewers agree that the paper addresses the timely and practical problem of efficient model routing. They recognize the modular predictor design of our work, which greatly reduces retraining overhead, enabling seamless, scalable routing for new models. `Reviewers NPzC and DE3V` also highlight the superior performance of combining general-purpose and learnable task-specific representations. `Reviewer NPzC` appreciate the framework's practical application in both constrained and unconstrained scenarios. All reviewers recognize the comprehensive evaluation that consistently shows improved accuracy-efficiency trade-offs over baselines.

`Reviewer d5bV` raises W1–W3.

| Summary of reviews | Summary of author response |
|--------------------|----------------------------|
| **W1:** FLOPs may not accurately capture the efficiency of quantized models. | **A1:** FLOPs can be replaced with other appropriate efficiency metrics (e.g., those suitable for quantized models). |
| **W2:** Concern about predictor expressivity as the number of strategies grows. | **A2:** Each strategy has its own dedicated predictor to prevent overfitting and ensure scalability. |
| **W3:** Difficulty with routing for open-ended tasks that lack a clear accuracy metric. | **A3:** Open-ended tasks can use alternative quality signals (e.g., LLM-as-a-judge). |

`Reviewer NPzC` raises W1-W2 and Q1-Q2.

| Summary of reviews | Summary of author response |
|--------------------|--------------------|
| **W1, Q1:** The robustness of our framework under different choices of the embedding model. | **A1:** We show that replacing the embedding model used in the paper with both smaller and larger alternative embedders yields comparable performance, suggesting that our framework is robust across a wide range of embedding sizes and qualities. |
| **W2:** How should practitioners select the accuracy-cost trade-off weight in unconstrained routing? | **A2:** The weight $w$ directly encodes the tradeoff between accuracy and cost. Pareto curves in Figures 3-6 in the paper enable users to choose $w$ based on their desired balance (e.g., computation limits, minimum accuracy requirements, or the “elbow’’ of the Pareto curve for their application). |
| **Q2:** How does CONCUR handle cold starts for new strategies without historical data? | **A3:** We clarify that our work focuses on improving training-based routing frameworks, and cold-start or training-free routing is a promising future work. |

`Reviewer dMBg` raises W1-W3.

| Summary of reviews | Summary of author response |
|--------------------|--------------------|
| **W1:** Unclear formal definitions and routing inputs/outputs. | **A1:** We clarify the formal constrained and unconstrained routing objectives, inputs, and outputs, and note that we will further improve the presentation. |
| **W2:** Novelty of the framework. | **A2:** We highlight that the novelty of our approach lies in its modular, per-strategy architecture and multi-representation design, which addresses the major limitations of previous routing methods and enhances both scalability and effectiveness. |
| **W3:** Lack of ablation studies. | **A3:** We present ablation results demonstrating that the different components of our framework contribute useful information, with their combination yielding the best performance. |

---

> ### Author Response · Authors · 2025-12-03
> **Rebuttal summary (continued)**
>
> `Reviewer DE3V` raises W1-W3.
>
> | Summary of reviews | Summary of author response |
> |--------------------|--------------------|
> | **W1:** Scalability of training two predictors per strategy. | **A1:** We note that modular predictors scale more efficiently than monolithic routers because they allow incremental updates and fully parallel training, which is reflected in our lower training times (Table 5 in the paper). |
> | **W2:** Lack of theoretical justification for why modular predictors generalize better, especially in OOD settings; optimization formulations are clear but not novel in theory. | **A2:** We explain that recent theory on modularity (Boopathy et al., 2024) shows why decomposing prediction across strategies reduces task dimensionality and improves generalization, supporting our empirical OOD results. We also clarify that the novelty lies in the modular architecture that leverages multiple representations. Our predict-then-optimize framework is chosen over end-to-end learning due to better flexibility and results. |
> | **W3:** How the predictors capture task-strategy interactions. | **A3:** We clarify that each predictor learns task-strategy interactions by combining general-purpose and task-specific representations of both input tasks and strategies within an MLP, enabling flexible modeling of nonlinear relationships. |
>
> Overall, we believe our responses address the reviewers’ concerns regarding clarity, scalability, theoretical grounding, and empirical validation. We hope these clarifications highlight the technical novelty, practical relevance, and strength of our work.

---

### Meta-Review · Area_Chair_7fmj · 2026-01-08

**Summary:**

The reviewers generally praised several core aspects of the paper
* The practically relevant and timely problem, and modular design solution leading to a very strong and promising practical approach
* Strong empirical performance of the method and thorough evaluation
* Supporting both constrained and unconstrained scenarios broadening applicability of the work

The core concerns that informed my suggested decision are:
1. Reviewer d5bV's questioning of the use of FLOPs to measure efficiency/cost of a strategy, and the more general question of not including models with different levels of quantization/precision as tradeoff-points
2. Some of the questions about additional ablations (NPzC, dMBg)
3. Questions about novelty and problem formulation/terminology (dMBg)
4. Computational overhead at inference/routing time due to using separate predictors for each strategy (DE3V)

**Reviewer Concerns:**

Concern 1 mentioned above was partially addressed in the rebuttal. The authors point out that FLOPs is fair for the strategies they consider since they do not use different precision. However, the reviewer raises an interesting point with this question, and this would be great to study further in future work, especially given the paper has focused on practical applicability of the approach.

The remaining concerns, 2--4, were all addressed to my satisfaction by the rebuttal.
2. Additional experiments were done and would be added to the paper to cover the ablations suggested.
3. This was largely an understanding and resolved/clarified in the rebuttal by pointing to the sections of the paper that covered those topics. Personally, I found the paper clearly written and accessible.
4. Tradeoffs are discussed and a reasonable argument is put forward for while the proposed approach is reasonable even if additional computation is incurred at routing time.

**Reviewer Scores:**

I do not pretend to be able to predict how specific reviewers would change their score if they had been able to participate fully in the discussion. The comments below reflect how I would hope they would have responded, based on my reading of the paper, the reviews, and the rebuttal.

Reviewer d5bV's score was already trending positive at 6 marginally above. The responses to their concerns were somewhat limited, so they may not have changed their score (being conservative).

Reviewer NPzC's score was also initially 6 marginally above. The rebuttal very clearly addressed the concerns they raised, so unless they felt strongly that concerns raised by other reviewers were still a remaining issue, I guess they would likely increase their score to solid accept.

Reviewer dMBg's score was initially 4 marginally below. The rebuttal very clearly addressed the concerns they had raised, and so I would expect they would also raise their score to at least marginally above.

Reviewer DE3V's score was initially 4 marginally below. All of the weaknesses they cited were satisfactorially addressed so I would expect they woudl also raise their score to at least marginally above.

Overall, this would lead to a clear accept score.

---

### Decision · Program_Chairs · 2026-01-26

Accept (Poster)